# Job Opening for Nucleosome Mechanic: Flexibility Required

**DOI:** 10.3390/cells9030580

**Published:** 2020-03-01

**Authors:** Mary Pitman, Daniël P. Melters, Yamini Dalal

**Affiliations:** 1Department of Chemistry and Biochemistry, Institute for Physical Science and Technology, University of Maryland, College Park, MD 20740, USA; mary.pitman@nih.gov; 2Chromatin Structure and Epigenetic Mechanisms, Laboratory of Receptor Biology and Gene Expression, Center for Cancer Research, NCI, NIH, Bethesda, MD 20892, USA; daniel.melters@nih.gov

**Keywords:** chromatin, mechanics, nucleosomes, dynamics, epigenetics

## Abstract

The nucleus has been studied for well over 100 years, and chromatin has been the intense focus of experiments for decades. In this review, we focus on an understudied aspect of chromatin biology, namely the chromatin fiber polymer’s mechanical properties. In recent years, innovative work deploying interdisciplinary approaches including computational modeling, in vitro manipulations of purified and native chromatin have resulted in deep mechanistic insights into how the mechanics of chromatin might contribute to its function. The picture that emerges is one of a nucleus that is shaped as much by external forces pressing down upon it, as internal forces pushing outwards from the chromatin. These properties may have evolved to afford the cell a dynamic and reversible force-induced communication highway which allows rapid coordination between external cues and internal genomic function.

## 1. Introduction

Chromosomes are complex nucleoprotein polymers, which contain a treasure trove of genetic and epigenetic data which is folded in the constrained space of the nucleus. The chromatin polymer permits spatiotemporal regulation to correctly access, repress, replicate, repair, and segregate the genome. Transcription of the genome is highly sensitive to external cues and stimuli, often bursting on a seconds time scale [1,2]. The base-unit of chromatin is the nucleosome, a core of eight histone proteins that wrap 147 bp of DNA, which forms a bead-on-a-string nucleosome array. Therefore, understanding the material properties of nucleosomes, and how such properties can be modulated, is an exciting new avenue in chromosome biology. At the nanoscale, macromolecular structures rely on elasticity, viscosity, and thermal motion for mechanical force development [3,4]. For example, chromosome movement during mitosis is dictated by frictional resistance [5,6,7,8]. These characteristics arise when the Reynolds number, which quantifies the relative importance of friction and inertia, is very low [9]. Thus, only when forces are generated, will there be action. The moment force production ceases, the action stops. Consequently, for a molecular motor to function, force generation is necessary. For instance, a wave-ratchet-wave model would allow for repeated force generation. Indeed, one mode by which chromatin remodelers are thought to function—in an ATP-dependent manner—is such a mechanism [10].

Physical properties of macromolecules that influence the amount of friction, such as elasticity, dictate how these macromolecules interact with each other. Structural proteins frequently have elastic mechanical properties [11] providing both strength and extensibility, together with low resilience, giving them notable resistance to fracture. Extreme examples are mammalian tendons and spider dragline silk [12,13]. Elasticity is formally defined by Hooke’s law, which states that the strain in a material is proportional to the applied stress within the elastic limit, which is quantified by Young’s modulus. Elastic properties of macromolecules are encoded by their quaternary and higher order structures, as either structurally disordered networks of flexible fibers, or structurally ordered macromolecules [14,15]. Chromatin fibers are in the regime of elastomers found elsewhere in nature, yet little is understood about how equivalent mechanical properties might contribute to its function. Deciphering how mechanical inputs alter the chromatin fiber, and conversely, how chromatin communicates cues back to the nuclear membrane, are relatively underexplored topics in biology. In this review, we highlight recent work on the material properties and dynamics of chromatin, and discuss how such properties might translate to chromatin fiber function in healthy states, which may be abrogated in the diseased state.

## 2. Nanoscale Dissections of Chromatin Mechanobiology

The elastic nature of chromosomes was first observed decades ago when the elastic recoil of chromatin was visualized by micro-manipulation experiments [16]. Indeed, the earliest documented quantitative micromanipulation experiments on intact cells were performed by Ethel Glancy D’Angelo in 1946 [17], when she reported that chromosomes from various *Chironomus* species can be elastically stretched to five times their original length without hysteresis [17,18]. A decade later, Bruce Nicklas used microneedle experiments coupled with radiation-induced damage of grasshopper chromosomes and reported similar results [19]. These early studies were made possible by increased spatial resolution of micromanipulators, and clearly demonstrated that the chromatin polymer had mechanical properties beyond that expected from simple folding of a linear chain.

New technological and theoretical developments continue to move questions of biological importance within reach. The development of micro-magnetic-tweezers, capable of manipulating chromatin on the nanometer scale, made it possible to measure the elasticity of biological molecules. A hierarchical decrease in Young’s modulus is observed for chromatin factors, where DNA is much more rigid compared to nucleosomes, which in turn are more rigid than chromosomes, whereas the nucleus has the lowest Young’s modulus (Table 1). It is important to keep in mind, however, that when discussing the material properties of biological mediums, context matters. Biological parameters such as charge, temperature, pressure, and cellular microenvironment can alter reported material properties.

## 3. Mechanical Signal Transduction into the Nucleus

The elastic properties of chromatin and the nuclear environment play a direct role in signal transduction by mechanical force, that is, mechano-transduction [33,34,35,36]. While the primary dogma in biochemistry is signal transduction through chemical pathways, mechanical signal transduction may be equally important with the benefit of a direct and rapid response. For example, while activation of the Src kinase by epidermal growth factor occurs over tens of seconds, mechano-transduction by integrins occurs in less than 0.3 s [33]. The nucleus is subjected to a variety of different mechanical forces, such as stretching, pushing, compression, tension, and shear stress [37], making it likely that the cell has evolved mechanisms to employ these signals to execute rapid, possibly transient, changes in gene expression programs.

The biochemical response to a mechanical signal for chromatin condensation and gene expression [38,39] has recently been explored by various labs as a rheostat of chromatin regulation [34,35,36,37,40,41,42,43]. Mechanical signaling also plays a vital role within the nucleus during cell division. For the faithful segregation of mitotic chromosomes, the spindle assembly checkpoint is regulated by pushing and pulling forces generated during mitosis. When a critical tension is reached, anaphase proceeds [8]. In this review we focus primarily on mechanical transactions that occur upon the chromatin fiber in the interphase stage of the cell cycle.

### 3.1. Mechanical Communication between the Cytoskeleton and Chromatin

The interphase nucleus receives mechanical inputs from inside and outside the cell via the cytoskeleton. For example, nuclear migration and nuclear positioning are critical during organismal developmental [44,45,46] and are coordinated by microtubules and actin dynamics (Figure 1). The cytoskeleton is connected to the nucleus via the linker of nucleoskeleton and cytoskeleton (LINC) complexes, which link the nucleoskeleton to the cytoskeleton, through a network of SUN and KASH domain containing proteins. The SUN domain spans the inner nuclear membrane, creating the physical connection with the nuclear lamina network, whereas the KASH domain spans the outer nuclear membrane. Different KASH domain proteins interact with either actin, intermediate filament, or microtubules. Interestingly, the length of the KASH domain corresponds to the amount of force that can be transferred [47]. Altering the length of the KASH domains resulted in defective nuclear migration and nuclear anchorage in the nematode *C. elegans*. This nucleoskeletal tethering has been found to be a common feature, especially for meiotic chromosomes [44,45,48].

In interphase cells, mechano-transduction mediated by cytoskeleton-tethering alters gene expression. For example, mechanical to chemical signal propagation in chromatin was shown to occur in a recent study in which magnetic beads were attached to the cellular surface to apply a controlled external force [38]. In another demonstration of mechano-transduction, chromatin deformation, and force-induced transcription of GFP tagged dihydrofolate reductase transgene was observed [38]. A third example is where the mobilization of local chromatin sites through a complex of transcription factors and myosin, utilizing the nuclear-actin matrix and heat shock protein Hsp90, resulted in the enrichment of the local chromatin with H2A.Z nucleosomes [49].

By measuring the root mean square fluctuations of the nuclear envelope in real time, Schreiner and colleagues found that untethered chromatin resulted in highly deformable nuclei [50]. Indeed, the integrity of the nuclear envelope is important to maintain nuclear stability under these forces [51]. Thus, it appears that a physical link between the cytoskeleton and chromatin provides a means by which the cell can protect and regulate the nucleus in a spatiotemporal manner.

### 3.2. Chromatin Deformation in Response to Cell Motility 

Cell migration is commonly found in organisms, for example, lymphocytes, macrophages, and metastatic cancer cells all migrate. Such cells squeeze through very small spaces, forcing the cellular interior to drastically deform (Figure 2). One mechanism that these cells deploy during this process is to condense their chromatin [34,36]. These kinds of chromatin distortions are fascinating to consider because they occur, presumably reversibly, on a relatively fast time scale. It is not yet known whether during these motility-induced chromatin conformational changes, transcribed chromatin is more susceptible to DNA damage or faulty DNA repair under force, and whether condensed chromatin associated with the nuclear lamina adopts a state which is protective. One could hypothesize that heterochromatin, which has increased local contacts with the nuclear periphery, may have a better buffering capacity to dissipate shearing forces. Indeed, evidence supports this notion, because DNA damage commonly correlates with local decondensed or relaxed chromatin [52,53], although some heterochromatin domains do not decondense upon DNA damage [54]. Concurrently, it will be fascinating to study how species which lack heterochromatin factors [55] adapt to shearing forces.

Mechanical loading of chromatin as a consequence of cell migration is dependent on the actin cytoskeleton and LINC complexes [59,60] (Figure 3A). Cytoskeletal contraction induces release of ATP and Ca^2+^, mediating the import of methyltransferases and histone deacetylates, both silencing and activating gene expression [34,35,40]. Nevertheless, how chromatin stretching precisely activates a specific genetic program remains unclear. Indeed, prolonged force application has been shown to result in decoupling of heterochromatin from the nuclear lamina, which results in extensive transcriptional silencing [61]. 

Based on these studies, an intriguing idea emerges that the chromatin fiber may function as a time-dependent, mechanosensitive rheostat. Subtle fluctuations in time and force scales can result in different outcomes at discrete locations on the fiber.

## 4. Mechanobiology of the Chromatin Fiber Across Scales

Only a fraction of the total chromatin is in direct contact with the nuclear periphery; indeed, the bulk of chromatin is located in the nucleoplasm. The shear abundance of nucleosomes and other nuclear proteins make them difficult to study at the single molecule resolution, in vivo. Major questions that have dominated the field for decades are whether the genome is organized in the nucleus, and whether such organization reflects function [37]. The development of chromosome conformation capture (Figure 3C) [62] sparked a massive push in the development of both proximity-ligation sequencing technologies and computational models to analyze and interpret the experimentally produced data [63,64]. How the three-dimensional organization of the genome, as well as distinct loci inform us about functional interaction is subject to intense studies. As discussed in detail below, to answer these fundamental questions, a synergistic confluence of in silico computational modeling and experimental biological approaches have been developed [62,65,66,67].

### 4.1. Computational Advances in Mechanobiology of Chromatin and Nucleosomes

To analyze and comprehend experimental data (reviewed in [68]), such as Hi-C, novel biophysical models were developed in an attempt to reconstruct chromosome topology and folding [34,35,36]. Statistical mechanics based chromatin models trained on Hi-C contact maps have also been developed, inspired by maximum entropy approaches previously applied to the protein folding problem [69,70]. Notable pioneering efforts resulted in applying an energy landscape model to extract the equilibrium ensemble of chromatin conformations known as the minimal chromatin model [71]. Interestingly, this minimal chromatin model shows a bias for open chromatin to localized at the edge of chromosome territories and phase separation between chromatin types. In addition, experimental data is also vital as a comparison for bottom-up approaches where a physical model is derived and then the output is checked for accuracy. One example of a recent top-down model [72] implemented the physics of block co-polymers and recapitulated *Drosophila* Hi-C data. Their tool, IC-Finder, predicts how 3D chromatin folding influences gene regulation. Such algorithms could also conceivably be developed for mechanical studies where chromatin topology is altered in response to force. DNA geometry and electrostatic interactions between nucleosomes can also inform chromatin models through twist angles between consecutive nucleosomes [73,74,75] or through steric studies that assume maximally compact face-to-face packing of the nucleosomes [76,77]. However, other approaches have historically leveraged the elasticity of chromatin for their predictive power. These methods can involve Brownian dynamics or Monte Carlo simulations [73]. Work was also done to model how nanoscale physical and geometric changes perturb the structure of chromatin at the mesoscale level and found that DNA elasticity, local geometry and linker DNA length play a key role in the topology of genome packaging [78].

Another major advantage of all-atom molecular dynamics (MD) simulations, is the capacity to observe minute fluctuation across an entire nucleosome, something single-molecule experiments cannot. Comparing the canonical H3 nucleosome to the centromere-specific CENP-A nucleosome showed that CENP-A nucleosomes display greater internal motions, in part because of the shearing motion at the four-helix bundle [79]. These results were extended by the recently developed minimal cylinder analysis (MCA) [80], which allows the Young’s modulus to be derived from all-atom MD simulations. One logical prediction of a macromolecule with greater intrinsic internal motions is that they are more elastic (lower Young’s modulus) than their canonical counterpart. Indeed, this is what was observed [15]. Interestingly, MCA showed that the heterotypic CENP-A:H3.3 nucleosome, which has been observed in cancer cells [81,82], had a Young’s modulus in between CENP-A and H3 nucleosomes [80]. Furthermore, the CENP-A binding protein CENP-C suppressed the internal motions, resulting in a higher Young’s modulus [15]. In addition to histone variants, the linker histone H1 family [83] impact the chromatin fiber dynamics, most notably by inducing chromatin condensation [84]. A recent course-gain mesoscale modeling study showed that different H1 variants differently alter the fiber structure, sedimentation rate, packing ratios, and bending propensities [85]. These data argue for a link between mechanical properties of nucleosomes and chromatosomes (nucleosomes + H1) and epigenetic control.

Another epigenetic layer are the post-translational modifications (PTMs) of nucleosomes, which have been found to alter nucleosome dynamics in silico. For instance, charge altering PTMs may increase transcription, nucleosome accessibility, and that additive effects of PTMs might be nonlinear [86]. MD simulations showed that acetylation of the core histone residue (K122 in H3 and K124 in CENP-A) rigidifies the nucleosomes in such a way that kinetochore seeding is blocked [87,88,89,90]. Intriguingly, MD simulations predicted the acetylation of K124 of the CENP-A nucleosomes to negatively affect CENP-C binding. Indeed, in vivo experiments confirmed these computational predictions [87]. 

Cumulatively, advances in computational modeling have incorporated biophysical principles to advance our understanding of how a long nucleoprotein polymer behaves. With cutting edge biophysical tools that can be applied in vitro and in vivo, these theoretical frameworks and resulting predictions can be rigorously tested by experimentation.

### 4.2. Experimental Advances in Mechanobiology of Chromatin and Nucleosomes

Although most of the nucleosomes found in the nucleus are canonical H3 nucleosomes, nucleosomes containing histone variants and PTMs tend to accumulate at certain sites, such as actively transcribing enhancers, promoters (Figure 3E), genes, sites of DNA damage (Figure 3D), centromeres (Figure 3C), and telomeres. This raises the question: How do pioneer and transcription factors find their sites of activity fast and efficiently in a sea of apparently uniform chromatin [91]? Do active sites stand out from the rest of the genome? One possibility that has been explored in recent years is that modified nucleosomal patches [92,93,94] create a unique local chromatin environment, simply by having distinct material properties that cause them to extrude locally from the rest of the genome. A simple analogy is to alter a stitch when knitting, which generates an identifiable pattern that pops out from the background. Taking this one step further, a “like-attracts-like” origami folding model can be envisioned, in which the three-dimensional folding of the genome contributes to the affinity between distant loci as a consequence of their chromatin fiber’s innate similar material or topological properties.

Examples of the extrusion idea, whether by mechanical features, or phase transitions, exist in the literature [15,80,95,96,97]. First, a recent cryoEM study of tri-nucleosomes showed that a single CENP-A nucleosome flanked by two H3 nucleosomes created an untwisted structure, demonstrating the power of a single nucleosome to alter the “stitch” of the fiber [95]. A second example comes from a recent study (from our lab), which dissected nucleosomal elasticity for the first time using both computational approaches and nanoindentation AFM studies. We reported that CENP-A nucleosomes are twice as elastic as canonical H3 nucleosomes, but that kinetochore protein binding suppressed the elastic state. Indeed, in vivo, a domain of more elastic CENP-A nucleosomes, in a bed of otherwise rigidified kinetochore bound CENP-A was postulated to extrude small stretches of elastic chromatin fiber, upon which the local recruitment of transcription machinery depended [15,80]. A third example is that of fission yeast heterochromatin protein Swi6, which binds to H3K9me3 nucleosomes, and subsequently compacts and transcriptionally represses the chromatin fiber (Figure 3B). Interestingly, this interaction also induces increased nucleosome core accessibility as measured by HDX-MS and NMR [96]. Using in vitro system, using an optical trap, both modified mononucleosomes and mononucleosomes containing a histone variant altered the time it took an RNA polymerase 2 to unwrap DNA from a nucleosome [97]. 

In vitro mononucleosome studies have provided invaluable insights into how nucleosomes behave. Yet, in the cell, nucleosomes exist exclusively as part of a chromatin fiber in the context of a very crowded nucleus. Indeed, live cell imaging combined with computational modeling showed that RNA polymerase 2 constrains the motions of single nucleosomes [98]. Similarly, nucleosomes in the context of heterochromatin are also less dynamic [99] (Figure 3B). Indeed, a recent computational study suggested a role for nucleosomes in the kinetic proofreading of activator-promoter recognition [100], tantalizingly pointing towards a model where mechanobiological features of individual nucleosomes drive enhancer-promoter interactions. Therefore, it is of immediate interest to understand how changes in nucleosome distortions will impact the nucleosome’s capacity to be modified by chromatin remodelers, whether they will promote or inhibit RNA polymerase processivity, and whether this changes the 3D spatial configuration of the chromatin fiber. In addition to mitosis (Figure 3C), replication also induces dramatic changes to the chromatin fiber (Figure 3F). How these changes alter the surrounding chromatin or the capacity of the cell to adequately respond to mechanical forces remains unexplored.

## 5. Future Perspectives: Mechanobiology of Chromatin in Disease

Changes in cell mechanics are associated with pathologies such as osteoarthritis, asthma, cancer, inflammation, and malaria [101,102]. Mechanobiological properties of the chromatin fiber are highly dependent on the molecular composition of the chromatin, most notably histone variants, PTMs, and chromatin binding factors. One of the most dramatic events a chromatin fiber can experience is a double strand break. Indeed, in cancer cells, higher levels of dsDNA breaks are commonly observed [103]. During the DNA damage response, chromatin commonly decondenses [52,53]. This feature of chromatin decondensation is not unique to the DNA damage response. In prostate cancer for instance, chromatin is markedly relaxed [104]. This is in part driven by SWI/SNF recruitment to heterochromatin, resulting in eviction of the polycomb complex in an ATP-dependent manner [105]. By evicting polycomb complexes, a more general open chromatin state is created. It is conceivable that subsequently histone variant nucleosomes are aberrantly accumulated on the chromatin fiber, changing the mechanobiological properties of the genome. Indeed, aberrant accumulation of histone variant nucleosomes has been observed for the centromeric CENP-A nucleosomes [81,82,106], including at cancer-associated translocation hotspots [82]. The importance of tight regulation of deposition of histone variant nucleosomes is highlighted in Floating-Harbor syndrome, where one of the two copies of the H2A.Z genes, which differs by three residues, is misincorporated, resulting in a distinctive craniofacial development [107]. Indeed, developmental incorporation of H2A.Z can differ between two closely related species [108], highlighting how precise spatiotemporal incorporation of distinct nucleosomes can be critical. It has to be noted that incorporation of H2A.Z nucleosomes requires chromatin remodelers, which are also a key component along with variants of the linker histone, in generating nucleosome repeat lengths [109,110]. Chromatin remodelers are frequently mutated in cancers [111]. How these mutants alter the mechanical properties of the chromatin fiber is an unexplored question. In addition, histone genes are frequently mutated in cancer [112,113]. One striking example is the H3K27M mutation which is found in about 80% of pediatric diffuse intrinsic pontine gliomas [114,115,116]. This mutation inhibits PRC2 activity, resulting in global reduction in H3K27me3 levels and increased proliferation [117,118]. How a single mutated allele among the many copies of histone genes can contribute to disease is only now beginning to be understood. Nevertheless, the paradigm that emerges from the sum of these data, is that tiny changes to the chromatin fiber can have significant clinical outcomes, whether by misincorporation, mislocalization, or attenuated remodeling. A recent in vitro study showed that the incorporation of H2A.Z nucleosomes at a nucleosome free region by the SWR chromatin remodeler is temperature sensitive [119], adding yet another layer to how a physical property can subtly alter the epigenetic landscape. Altogether, we speculate that seemingly nanoscale changes alter the mechanical properties of chromatin in a context-dependent manner. Completely lacking from the current debate is also the evolutionary context in which these properties evolved. It will be of great interest to study whether mechanical properties or mechanotransduction pathways of chromatin are conserved, or diverge across the species, and whether such adaptive strategies co-evolved with discrete biological outcomes. These currently relatively understudied themes in chromatin biology present exciting future pathways for theoretical and experimental dissection using a broad array of cutting-edge technologies.

## Figures and Tables

**Figure 1 cells-09-00580-f001:**
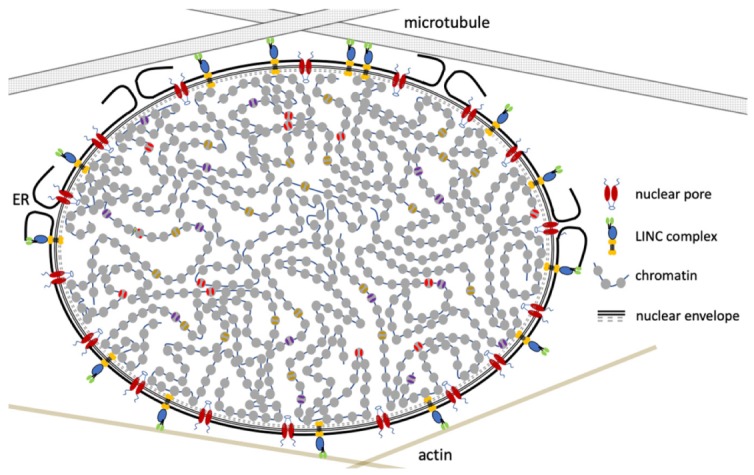
The nucleus is an integral part of the cell’s cytoplasm. The cytoskeleton interacts with chromatin through the linker of nucleoskeleton and cytoskeleton (LINC) complex, allowing mechanotransduction of signals from outside the cell to swiftly and directly impact chromatin structure and gene expression.

**Figure 2 cells-09-00580-f002:**
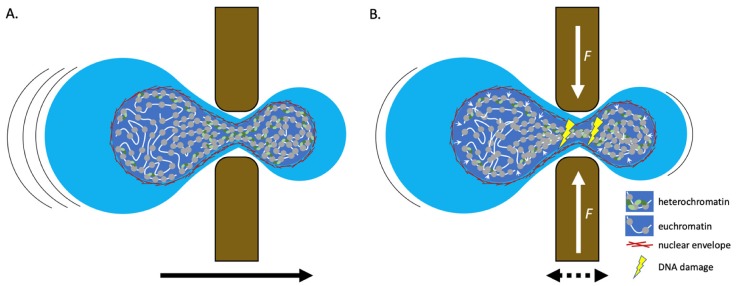
Chromatin dynamics during migration. (**A**) When a cell migrates (black arrows) through a small space, resulting in extensive compression of the cellular interior, chromatin tends to become hyper condensed. As we predict that heterochromatin is more gel-like than euchromatin, this might facilitate the nucleus to move through the gap faster, in a capillary suction-like manner. (**B**) When a cell is exposed to extensive compression forces (white arrows) or is stuck in a gap, the chromatin will eventually detach from the nuclear envelope and becomes more susceptible to DNA damage. This supports a model where chromatin-nuclear envelope contact serves to protect the DNA in a multi-contact manner. Intriguingly, dual functions for DNA repair factors have been observed. In addition to facilitating DNA repair, some DNA-repair factors have non-DNA repair functions. ATR that is localized to the nuclear periphery relocates upon osmotic stress and mechanical stretching, regulating chromatin condensation and nuclear envelope breakdown [56,57,58]. A deep mechanistic dissection of how DNA damage is repaired under force regimens will be extremely important, as well as how the DNA repair machinery manages mechanical forces.

**Figure 3 cells-09-00580-f003:**
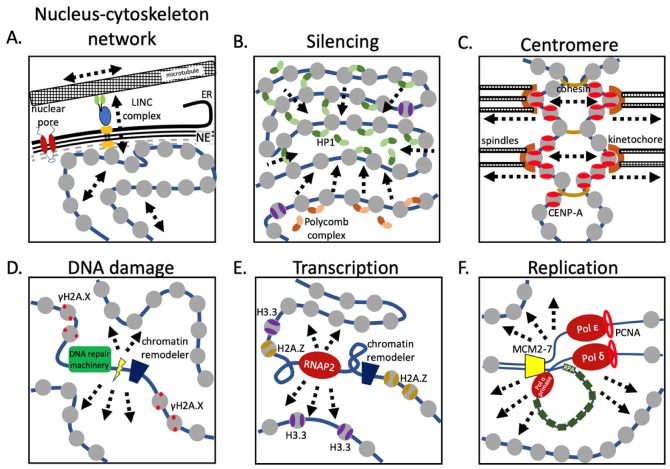
Nuclear machinery alter chromatin properties. (**A**) Physical forces are transmitted to the chromatin through a trans-nuclear envelope protein complex, altering both chromatin compaction and gene transcription. (**B**) Gene repression is commonly achieved through heterochromatin formation by either HP1 or Polycomb complexes, resulting in chromatin condensation and reduced chromatin elasticity. (**C**) Mitosis is a highly physical event, where in a highly coordinated manner, chromosomes are moved over extensive distances through the mitotic spindle machinery. The pushing and pulling forces that are excepted on the chromosomes, and especially the centromeric chromatin, are critical in satisfying the spindle checkpoint. (**D**) DNA damage must be repaired as quickly as possible, recruiting the DNA repair machinery, and locally decondensing chromatin, temporarily altering the local chromatin dynamics in part by enrichment of phosphorylated H2A.X nucleosomes. (**E**) Gene transcription is one of the most profound events in the nucleus, where the transcriptional machinery distorts the DNA double helix both behind and in front of the RNA polymerase. Highly transcribed regions are also enriched for H2A.Z and H3.3 nucleosomes. These nucleosomes might alter the local chromatin properties, potentially bringing together distant sites, such as enhancers and promoters. (**F**) Replicating chromatin has to decondense for the replication machinery to pass through, while at the same time maintaining distinct chromatin marks to be present on both newly formed strands. This highly dynamic and local event might temporarily increase the chromatin’s elasticity. ER = endoplasmic reticulum, NE = nuclear envelope.

**Table 1 cells-09-00580-t001:** The Young’s modulus across single molecules and macromolecules differ by orders of magnitude.

Substrate	Young’s Modulus	References
DNA	0.3–1 GPa	[20]
Protein	0.02–0.8 GPa	[21,22,23,24,25,26,27,28,29]
Nucleosome	2.8–15.4 MPa	[15]
Chromosome	40–400 Pa	[30,31]
Nucleus	250 Pa	[32]

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
