# Peer review of "Job Opening for Nucleosome Mechanic: Flexibility Required"

_cells, 2020, doi:10.3390/cells9030580_

Round 1

Reviewer 1 Report

The review by Pitman et al., entitled 'Job Opening for Nucleosome Mechanic: Flexibility Required' is a well written and timely contribution to the field. I only have minor comments:

lines 222-229: while the idea of distinct material properties is an exciting one, there should be clear references made in this paragraph to literature providing evidence for this theory   233-237: there should be a reference given for the work described earlier to avoid confusion   Outlook: the authors should highlight more that the paragraph on histone remodellers and histone mutants with respect to the biochemical properties of chromatin is (at the moment) speculation, but should also highlight work that showed how these mutations e.g. impair enzymatic reactions (in case e.g. H3K27M and Ezh2)   Some typos:

line 163: ‘workers’ is a strange expression

line 172: ‘a bias for open chromatin to BE localised’

Author Response

We thank you for the positive comments! We have incorporated all your suggestions as noted below.

Reviewer 1

The review by Pitman et al., entitled 'Job Opening for Nucleosome Mechanic: Flexibility Required' is a well written and timely contribution to the field. I only have minor comments:

lines 222-229: while the idea of distinct material properties is an exciting one, there should be clear references made in this paragraph to literature providing evidence for this theory  

We have added references as the reviewer suggested.

233-237: there should be a reference given for the work described earlier to avoid confusion  

We have updated the references as suggested.

Outlook: the authors should highlight more that the paragraph on histone remodellers and histone mutants with respect to the biochemical properties of chromatin is (at the moment) speculation, but should also highlight work that showed how these mutations e.g. impair enzymatic reactions (in case e.g. H3K27M and Ezh2)  

Thanks, this is a good suggestion. We have incorporated the interplay between histone mutations and impaired enzymatic reactions in the last section.

Some typos:

line 163: ‘workers’ is a strange expression

It is a bit old-fashioned, we agree. The sentence has been updated to remove the term ‘workers’. 

line 172: ‘a bias for open chromatin to BE localised’

Thanks, the typo has been fixed.

Reviewer 2 Report

This is a well written and succinctly covers the area of chromatic dynamics. It highlights key mechanisms that have added a lot to understand the mechanism of flexibility in chromatin.  I especially appreciate the highlights of what is not known and the future questions that need to be addressed.

A few minor comments are:

  • Authors have said heterochromatin might help to increase contact, what about cell or organism that lack heterochromatin. Do we have studies on these type cells or organism? How they execute the chromatin dynamics.

  • It might be helpful to have a table in section 4.1 to list the studies which have used new techniques to understand chromatin topology.

Author Response

We thank the reviewer for their positive comments, and have incorporated all their suggestions as noted below.

Reviewer 2

This is a well written and succinctly covers the area of chromatic dynamics. It highlights key mechanisms that have added a lot to understand the mechanism of flexibility in chromatin.  I especially appreciate the highlights of what is not known and the future questions that need to be addressed.

A few minor comments are:

Authors have said heterochromatin might help to increase contact, what about cell or organism that lack heterochromatin. Do we have studies on these type cells or organism? How they execute the chromatin dynamics.

The reviewer makes an excellent point. Budding yeast is a species that lacks many heterochromatin factors (or DNA methylation). It is known to physically anchor the telomeres to the nuclear periphery. To the best of our knowledge, no distinct studies have been conducted on how budding yeast chromatin responds to forces such as seen for human cells that are forced through narrow spaces. It is to be hoped this review will encourage yeast biologists to tackle these sorts of questions in the best studied model organism we have for chromatin. The only work we know of in budding yeast is from Kerry Bloom’s lab, who has done beautiful work elucidating the spring like behavior of centromeres using live microscopy (we cite a review by him on this topic). We have updated the text to reflect the current understanding for species that lack heterochromatin factors. We added a sentence at the very end of the Discussion highlighting the need to study these features across the species. 

It might be helpful to have a table in section 4.1 to list the studies which have used new techniques to understand chromatin topology.

Good point. A recent review in Molecular Cell by McCord, Kaplan, and Giorgetti summarizes exactly what the reviewer is asking for. We have included this reference for completeness.

Reviewer 3 Report

In the review article `Job Opening for Nucleosome Mechanic: Flexibility Required’ the authors Mary Pitman et al. provide a comprehensive overview of past and current findings on the mechanical properties of chromatin with an emphasis on nucleosome properties. Overall the review is nicely written, also allowing non expert readers to get an overview of the topic and still providing a detailed summary of recent findings in the field.

Minor comments

The properties of the LINC complex could be described a little bit more in detail

Table 1 lists the Young’s Modulus of several chromatin/nuclear substrates, but only the Young’s Modulus for the Nucleus is discussed in the text. The authors should also referee to the other values and describe the differences

Page 3 Line 99: “…positioning are an critical factor…” should read “a critical factor”

I am missing in this review some of the work of Marco Foiani, who contributed in my opinion to the understanding of linking mechanical properties of chromatin, nuclear envelop and DNA damage response (see, e.g. PMID: 27283761, PMID: 26008788, PMID: 25083873)

Page 5 Line 147 “…Indeed, prolonged force application has been show to…” should read “been shown to”

Figure 3A, B and F are not referenced in the text nor discussed in detail.

Figure 4 B is referenced in the text, but there is no figure 4

In vitro / in vivo are inconsistently written in italic throughout the text

Page 8 Line 296 The abbreviation “NRL” is used for the first time, but not defined.

Author Response

We thank the reviewer for their very positive and constructive comments.

We have incorporated all their suggestions as noted below in green. 

Reviewer 3

In the review article `Job Opening for Nucleosome Mechanic: Flexibility Required’ the authors Mary Pitman et al. provide a comprehensive overview of past and current findings on the mechanical properties of chromatin with an emphasis on nucleosome properties. Overall the review is nicely written, also allowing non expert readers to get an overview of the topic and still providing a detailed summary of recent findings in the field.

Minor comments

The properties of the LINC complex could be described a little bit more in detail

Thanks, we have expanded the properties of LINC as suggested.

Table 1 lists the Young’s Modulus of several chromatin/nuclear substrates, but only the Young’s Modulus for the Nucleus is discussed in the text. The authors should also referee to the other values and describe the differences

Thanks, we have updated the text to incorporate a complete discussion of Table 1 and Figure 3.

Page 3 Line 99: “…positioning are an critical factor…” should read “a critical factor”

Thanks for catching this, we have corrected the typo.

I am missing in this review some of the work of Marco Foiani, who contributed in my opinion to the understanding of linking mechanical properties of chromatin, nuclear envelop and DNA damage response (see, e.g. PMID: 27283761, PMID: 26008788, PMID: 25083873)

A very good point. We are sorry we missed these critical references, we now incorporate them.

Page 5 Line 147 “…Indeed, prolonged force application has been show to…” should read “been shown to”

Thanks, we have corrected the typo.

Figure 3A, B and F are not referenced in the text nor discussed in detail.

Thanks, we have updated the text as mentioned above.

Figure 4 B is referenced in the text, but there is no figure 4

We apologize for the error and have corrected this.

In vitro / in vivo are inconsistently written in italic throughout the text

Thanks, we have updated the text to exclusively use in vitro / in vivo in italic.

Page 8 Line 296 The abbreviation “NRL” is used for the first time, but not defined.

Apologies, NRL has been written out, it is nucleosome repeat length, the distinctive spacing between nucleosomes, which on average, is unique to each species.